# Transcriptome Analysis Reveals Critical Genes Involved in the Response of *Stropharia rugosoannulata* to High Temperature and Drought Stress

**DOI:** 10.3390/cimb47100835

**Published:** 2025-10-10

**Authors:** Shengze Yan, Shengyou Wang, Meirong Zhan, Xianxin Huang, Ting Xie, Ruijuan Wang, Huan Lu, Qingqing Luo, Wei Ye

**Affiliations:** 1Fujian Key Laboratory of Crop Genetic Improvement and Innovative Utilization for Mountain Area, Institute of Edible Fungi, Sanming Academy of Agricultural Sciences, Sanming 365500, China; yanshengze1102025@163.com (S.Y.); smwsy2022@163.com (S.W.); zhmrnhkl@163.com (M.Z.); hxianxin96@163.com (X.H.); 19376776419@163.com (T.X.); luoqingqing6@126.com (Q.L.); 2National Research Center for Edible Fungi Biotechnology and Engineering, Key Laboratory of Applied Mycological Resources and Utilization, Shanghai Academy of Agricultural Sciences, Shanghai 201403, China; wangruijuan@saas.sh.cn (R.W.); luhuan@saas.sh.cn (H.L.); 3Ministry of Agriculture, Shanghai Key Laboratory of Agricultural Genetics and Breeding, Institute of Edible Fungi, Shanghai Academy of Agricultural Sciences, Shanghai 201403, China

**Keywords:** *Stropharia rugosoannulata*, abiotic stress, DEG, candidate genes, transcriptome analysis

## Abstract

In this study, the differences in gene expression of *Stropharia rugosoannulata* at different treatment times under high temperature and drought stress were analyzed by transcriptomics. Here, a total of 74,571 transcripts and 16,233 unigenes were identified, with an average assembly length of 3002 bp. A total of 10,248 differentially expressed genes (DEGs) were identified. DEG analysis indicated that the numbers of DEGs under high-temperature stress for 1 d, 2 d, and 3 d were 798, 851, and 1484, respectively. These DEGs were involved in 96 GO functional categories and 69 KEGG metabolic pathways. Meanwhile, the numbers of DEGs under drought stress for 3 d, 6 d, and 9 d were 421, 1072, and 2880, respectively. These DEGs were involved in 108 GO functional categories and 78 KEGG metabolic pathways. Further analysis of the metabolic pathway (ko04011) commonly enriched by DEGs identified 15 candidate genes responding to high-temperature or drought stress. Eight candidate genes were randomly selected for qRT-PCR verification, and the qRT-PCR results were basically consistent with the transcriptome datasets. These findings provide critical candidate genes for understanding the molecular regulation mechanism of *S. rugosoannulata* in response to high temperature and drought stress and have important reference value for its stress resistance breeding.

## 1. Introduction

*Stropharia rugosoannulata* Farl. Ex Murrill is a kind of edible mushroom with high nutritional, low-fat, and medicinal values. It was found that the crude protein content of *S. rugosoannulata* was 25.75%, which was higher than the protein content of most edible fungi [1]. The total amino acid content ranges from 18.89% to 31.01% in recent reports, comprising 18 to 20 kinds of amino acids, with a crude fiber content of 13.3 g/100 g fruiting body [2,3]. The contents of phosphorus and potassium in the mineral elements were higher, at 3.48% and 0.82%, respectively [4,5,6]. *S. rugosoannulata* is also an extremely precious medicinal fungus, which has the effect of treating or improving many diseases in humans. In addition, *S. rugosoannulata* is one of the cultivated varieties of edible fungi recommended by the FAO, which offers good economic, ecological, and social benefits [1,5,7].

The growth and development of *S. rugosoannulata* require suitable humidity, temperature, light, pH, and air, with humidity and temperature being the most important factors [8,9,10]. When fungi are exposed to high temperatures and other adverse conditions, reactive oxygen species (ROS) accumulate in large quantities, leading to membrane peroxidation, damage to the cell membrane structure, and destruction of normal metabolism [11,12]. Under high-temperature stress, reactive oxygen species such as intracellular superoxide radical (O^2−^), hydrogen peroxide (H_2_O_2_), and hydroxyl (OH-) accumulate rapidly, which leads to membrane lipid peroxidation, which damages membrane lipids, nucleic acids, and membrane proteins, and finally leads to cell death [13]. High temperatures will reduce the quality and yield of fruiting bodies, making them susceptible to infection by various bacteria. Therefore, the tolerance of edible fungi to high temperatures is very important. High temperature will affect the mycelium stage, primordial stage, and fruiting stage of edible fungi to varying degrees [14,15]. In the mycelium stage, high temperature will change its morphology, affect its growth rate, make its biomass change significantly, and form an obvious heat shock circle. The primordial stage is very sensitive to changes in the external environment. High-temperature stress slows primordial differentiation, resulting in a low differentiation rate, and even mushroom growth is hindered, making it very easy to cause fungus bag pollution [16,17]. If the temperature is too high during the fruiting period, the fruiting body will be deformed and not fruit, which will lead to a decline in yield and even no harvest. Understanding the molecular mechanism of edible fungi responding to heat stress can lay a foundation for cultivating high-temperature varieties of edible fungi [10,11]. The effects of high temperature on macrofungi have been reported in *Pleurotus ostreatus* [18], *Pleurotus eryngii* [19], *Agaricus bisporus* [20], and *Ganoderma lucidum* [21]. With the aggravation of land drought, it is necessary to maintain soil water content and air relative humidity by increasing water consumption and water use times to cultivate *S. rugosoannulata* [22]. According to national standards, when the relative water content of soil is lower than 50%, it belongs to severe soil drought. For example, when the soil is subject to severe drought, it will lead to the slow growth of *Morchella esculenta* and the decline of fruiting body quality and yield [11,23].

In recent years, with the rapid development of high-throughput sequencing technology and the substantial reduction in sequencing cost, high-throughput sequencing has become the first choice for genome and transcriptome analysis of plants, animals, and edible fungi. Increasingly, reports have investigated the transcription and expression of genes in edible fungi under abiotic stress from an omics perspective, aiming to elucidate the regulatory mechanisms of physiological processes related to edible fungi [24]. Transcriptomics can study the transcriptional regulation of genes in cells at an overall level and reveal the molecular mechanisms of complex biological pathways and trait regulation networks. Exploring the transcriptomics changes in edible fungi under adversity stress is helpful to deeply understand the complex molecular regulatory network of edible fungi under adversity stress and provide a reference for improving the stress resistance of edible fungi [24]. Kim et al. [25] used RNA-seq to compare the morphological changes and gene expression of *Lentinula edodes* fruiting bodies under blue light and dark conditions. The results showed that blue light mainly induced cap growth, while dark cultivation promoted stalk length development, and a total of 221 DEGs were significantly up-regulated and 541 DEGs were significantly down-regulated, and 8 up-regulated genes under blue light conditions were identified, such as *DDR48* (heat shock protein), Fasciclin-domain-containing protein, carbohydrate esterase family 4 protein, and FAD NAD-binding domain-containing protein that are involved in morphological development of primordium and embryonic muscle development, cell adhesion, and affect the structure of cellulose. Yu et al. [26] studied the response pattern of *L. edodes* to heat stress by transcriptome and found that heat-tolerant strains showed more DEGs at high temperature, and two DEGs were highly expressed at all developmental stages. In addition, several possible candidate genes involved in the Cd accumulation were identified, including the major facilitator superfamily genes, heat shock proteins, and laccase 11, a multicopper oxidase. Hao et al. [27] performed an RNA-seq analysis of *S. rugosoannulata* strain DQ1 (high-sensitivity) and *S. rugosoannulata* strain DQ3 (low-sensitivity) under low-temperature stress, and finally, a total of 9499 DEGs were screened. Additionally, further research showed that carbohydrate enzyme (*GT*, *CE*, *GH*, and *AA*) genes were down-regulated more significantly in DQ-1 than DQ-3, and several cellulase activities were also reduced to a greater extent. Moreover, the *GR*, *POD*, *CAT2*, and *CAT1* genes and more heat shock protein genes (*HSP78* and *HSP20*) were up-regulated in the two strains after low-temperature stress, and the *GPX* gene and more heat shock protein genes were up-regulated in DQ-3. At present, there is no report on analyzing the gene expression and identifying candidate genes of *S. rugosoannulata* in response to drought and high-temperature stress via transcriptome sequencing.

In this study, *S. rugosoannulata* strain D-3 was selected as the experimental material, and three treatments of normal, high temperature, and drought stress were conducted. Transcriptome sequencing was used, and bioinformatics analysis was carried out. The gene expression of *S. rugosoannulata* in response to stress was detected from the mRNA level, the candidate genes closely related to drought and high-temperature stress were screened, and the expression patterns of randomly selected genes were verified. In conclusion, these findings aim to provide a theoretical basis for stress resistance regulation and breeding of *S. rugosoannulata*.

## 2. Materials and Methods

### 2.1. Samples and Stress Treatments

In this study, the *S. rugosoannulata* strain D-3 was derived from the strain preservation center of Sanming Academy of Agricultural Sciences and was used as research material to study its stress resistance mechanism. For the control and drought stress and high-temperature stress treatments, the fermented cultivation materials were placed in the mushroom incubator, the temperature was set to 20–25 °C, and the carbon dioxide concentration was 1000–1500 ppm. Next, 35–45 d after sowing, the temperature was adjusted to 13–15 °C, and after 7 d, the mycelium began to kink and produce mushrooms. Then, the illumination time was set to 6 h/d, the humidity was 80–90% and 60–70%, respectively, and the temperature and humidity detector was used to ensure that the water content of soil and cultivation material was 55–60% and 40–45%, respectively. The drought stress treatment was carried out for 3 d, 6 d, and 9 d. In addition, during the fruiting period, *S. rugosoannulata* (fruiting body diameter of 1–2 mm) with the same growth vigor were placed in the fruiting incubator at 15 °C (control) and 35 °C (stress treatment), respectively, for 1 d, 2 d, and 3 d of high-temperature stress treatment. After harvesting all the treated *S. rugosoannulata*, mushrooms with the same growth vigor (three replicates) were randomly selected and quickly placed in liquid nitrogen for freezing, then stored at −80 °C in a freezer for later use.

### 2.2. Total RNA Extraction, Construction of the cDNA Library, and RNA-seq

The total RNA of all samples was extracted by Trizol (Invitrogen, Carlsbad, CA, USA), and the genomic DNA residue was removed by DNase I, and the purity and concentration of the extracted total RNA were determined by a NanoDrop One (Thermo Fisher Sci, Waltham, MA, USA) and an Agilent 2100 Bioanalyzer (Agilent Technologies, Waldbronn, Germany). All qualified RNA samples were sent to Biomarker Technologies Co., Ltd. (Beijing, China) for cDNA library construction and transcriptome sequencing. According to the manufacturer’s instructions, cDNA libraries were constructed using the TruSeq RNA sample preparation kit (Illumina RS-122-2101, Illumina, CA, USA). Then, all the acceptable cDNA libraries were sequenced on an Illumina HiSeqTM 3500 sequencing platform. The clean reads were processed by data filtering, and low-quality reads were discarded. The RNA-seq raw sequence datasets were deposited in the National Genomics Data Center (NGDC), China National Center for Bioinformation, Chinese Academy of Sciences (CRA028883) (https://ngdc.cncb.ac.cn/gsa/browse/CRA028883, released on 19 August 2025, accessed on 11 August 2025). In the NCBI database, the genome of Stropharia rugosoannulata strain QGU27 (GCA_028532985.1) has been posted (https://www.ncbi.nlm.nih.gov/datasets/genome/GCA_028532985.1/, accessed on 21 September 2025). Considering that large differences between the sequencing data and the reference genome may occur, which will affect the alignment results, we use Trinity v2.15.1 (California, USA) [28] to splice clean reads to obtain transcripts and then carry out subsequent analysis. According to the expression level of genes in different samples, DEG analysis, functional annotation, and functional enrichment of DEG analysis were carried out to explore the critical candidate genes involved in the response to abiotic stress and their regulatory pathways in *S. rugosoannulata*.

### 2.3. Discovery of New Genes and Functional Annotation

Based on the transcriptome data, the software Trinity v2.15.1 (California, USA) [28] is used to construct reference transcripts and discover new transcripts or unigenes of this species. Based on the basic biological principle that structure determines function, the homologous sequence with the highest similarity to the new gene sequence in each gene function annotation database is found by sequence comparison, and the gene function annotation information of the homologous sequence is used as the functional description of the new gene discovered. Based on NR (ftp://ftp.ncbi.nih.gov/blast/db/, accessed on 5 July 2025), Pfam (http://pfam.xfam.org/, accessed on 5 July 2025), PlantTFDB (http://plntfdb.bio.uni-potsdam.de/v3.0/, accessed on 5 July 2025), GO (http://geneontology.org/, accessed on 5 July 2025), KOG (https://www.ncbi.nlm.nih.gov/research/cog-project/, accessed on 5 July 2025), KEGG (https://www.genome.jp/kegg/kegg1.html, accessed on 5 July 2025), and other databases, we used BLAST 2.9.0+ (NCBI, Bethesda, USA) [29] to annotate the new genes in depth and obtain their annotation information.

### 2.4. Differential Expression Gene Analysis

After comparing the sequence of clean reads with the assembled reference transcript to obtain the corresponding position information, the expression level of the transcript and the gene can be quantified by the position information of mapped reads on the gene. In order to make the number of fragments truly reflect the transcript expression level, it is necessary to normalize the number of mapped reads and the transcript length in the sample and adopt FPKM (Fragments per kilobase of transcript per million fragments mapped) as an index to measure the transcript or gene expression level. In the process of detecting DEG, the log_2_FC (fold change) ≥ 1 and the false discovery rate (FDR) < 0.05 are used as screening criteria. The Benjamini–Hochberg correction method is adopted to correct the significant *p*-value obtained from the original hypothesis test. Subsequently, EBSeq (Madison, WI, USA) [30] was used for differential analysis to obtain a set of DEGs between the two samples. In addition, hierarchical clustering analysis was performed on the screened DEG, and the genes with the same or similar expression patterns were clustered, and the *K*-means method was used for statistical and clustering analysis of the DEG.

### 2.5. Enrichment Analysis of GO Function and KEGG Metabolic Pathway

The enrichment analysis of GO and KEGG metabolic pathways of DEGs was performed by the R package clusterProfiler Version 4.1.1 (Nanjing, China) [31], and both were accurately tested by Fisher. After correction, GO entries and KEGG metabolic pathways with *p* < 0.05 were considered to be significantly enriched.

### 2.6. Screening of Critical Differentially Expressed Genes and qRT-PCR Analysis

According to the results of KEGG enrichment analysis, the DEGs enriched in the same metabolic pathway were further annotated to screen candidate genes in response to high temperature and drought stress. Based on the identified candidate genes, eight genes were randomly selected to analyze their expression patterns in all samples by qRT-PCR. The first strand cDNA synthesis was carried out according to the protocols of the Prime-ScriptTM RT Reagent Kit (TaKaRa, Shimogyo-ku, Kyoto, Japan). The glyceraldehyde-3-phosphate dehydrogenase (*GAPDH*) was chosen as an internal control gene to normalize the expression data. The eight primer pairs were designed by the software Primer Premier 5.0 according to the coding sequences (CDS) of the selected candidate genes, and primer details are provided in Appendix A. The qRT-PCR amplification and procedure were carried out according to the report of Cheng et al. [32]. Three biological replicates for each sample were performed, and the relative expression level of each selected candidate gene in different stress treatments was calculated from the 2^−∆∆Ct^ value [33]. Histograms and the significance of the difference in expression levels between different samples were analyzed by the software GraphPad Prism 8.

## 3. Results

### 3.1. Summary of the Transcriptome Sequencing and Data Processing

In our study, 21 cDNA libraries were established, including six stress treatment groups (gh3d, which means response to drought for three days; gh6d, which means response to drought for six days; gh9d, which means response to drought for nine days; gw1d, which means response to high temperature for a day; gw2d, which means response to high temperature for two days; and gw3d, which means response to high temperature for three days) and a control group (ck) with three replicates in each group. RNA-seq generated 41.96 million to 49.15 million raw reads for each library (Table 1). Subsequently, after removing the low-quality raw reads and adaptors, a total of 142.43 G clean reads were produced, and the average number of clean reads for the gh3d, gh6d, gh9d, gw1d, gw2d, gw3d, and CK treatments was 6.78, 6.58, 7.12, 6.83, 6.36, 6.83, and 6.97 Gb, respectively. The percentage of bases with Q30 was more than 97.00% in each group (Table 1). The above results suggest that the RNA-seq data are reliable and can meet the requirements of subsequent transcriptome bioinformatics analysis.

### 3.2. Transcriptome Sequence Assembly and Gene Function Annotation

The sequences with more than 99% similarity were merged and assembled by Trinity software for transcriptome assembly, and the parameters were set to the default parameters. Finally, 74,571 transcripts were generated, with an average length of 3002 bp. The smallest transcript sequence length was 190 bp, and the number of transcripts longer than 2000 bp was the largest, accounting for 56.41% of the total. The N50 was 4550 bp, and the GC content was 49.74% (Appendix A). Subsequently, the filtered sequences were aligned and annotated in databases such as KEGG, KOG, NR, GO, and Pfam. The results showed that the number of unigenes successfully annotated in NR, KEGG, KOG, GO, and Pfam databases accounted for 55.11% (8591), 19.49% (3039), 30.21% (4709), 23.91% (3728), and 54.85% (8550) of the total number of genes, respectively. 65.74% (10,248) of the transcript sequences were successfully annotated in at least one database (Appendix A).

### 3.3. Sample Correlation Analysis and Screening of Differentially Expressed Genes

To reflect the correlation of gene expression among all samples, Pearson correlation coefficients of all gene expression between every two samples were calculated, and these coefficients were reflected in the form of heat maps. As shown in Appendix A, the repeatability and correlation of all the samples were high, and the subsequent analysis was highly reliable. The screening results of DEGs between ck vs. gw1d, ck vs. gw2d, and ck vs. gw3d showed that *S. rugosoannulata* differentially expressed 798 genes after 1 d of 35 °C high-temperature stress (gw1d), including 494 up-regulated and 304 down-regulated genes (Figure 1A). After 2 d of high-temperature stress at 35 °C (gw2d), a total of 851 genes were differentially expressed, and 548 and 303 genes were up-regulated and down-regulated, respectively (Figure 1B). After 3 d of high-temperature stress at 35 °C (gw3d), the total number of DEGs in *S. rugosoannulata* was 1484, including 963 up-regulated and 521 down-regulated genes (Figure 1C). With the prolongation of high-temperature stress time, the total number of DEGs in *S. rugosoannulata* gradually increased. In addition, the screening results of DEGs between ck vs. gh3d, ck vs. gh6d, and ck vs. gh9d showed that after 3 d of drought stress (gh3d), *S. rugosoanulata* differentially expressed a total of 421 genes, including 269 up-regulated and 152 down-regulated genes (Figure 2A). After 6 d of drought stress (gh6d), a total of 1072 genes were differentially expressed, with 711 up-regulated and 361 down-regulated genes, respectively (Figure 2B). After 9 d of drought stress (gh9d), the total number of DEGs in *S. rugosoanulata* was 2880, including 1501 up-regulated and 1379 down-regulated genes (Figure 2C). Clearly, as the duration of high temperature and drought stress prolongs, the total number of DEGs in *S. rugosoanulata* gradually increases. These findings suggest that as the duration of high temperature and drought stress increases, *S. rugosoanulata* will resist external adversity by regulating more genes.

### 3.4. GO Enrichment Analysis of Differentially Expressed Genes

GO enrichment analysis was conducted on the DEGs screened between the group of high-temperature and drought-stress treatment versus the control group. Under high-temperature stress, 212 DEGs were enriched in the ck vs. gw1d group (Figure 3A). In terms of cellular components (CCs), membrane, nucleus, fungal-type cell wall, and other related DEGs were significantly enriched. In terms of molecular function (MF), ATP binding, protein binding, oxidoreductase activity, flavin adenine dinucleotide binding, catalytic activity, and other related DEGs were significantly enriched. After high-temperature stress, obsolete oxidation-reduction processes, transmembrane transport, biosynthetic processes, and other processes play an important role in biological processes (BPs). A total of 214 DEGs were enriched by GO in the ck vs. gw2d group, and the DEGs were significantly enriched in transmembrane transport, obsolete oxidation-reduction process, response to stress, and other items in BPs. In CCs, differential genes were significantly enriched in the membrane, nucleus, and other items. In MF, differential genes were significantly enriched in oxidoreductase activity, ATP binding, heme binding, and catalytic activity (Figure 3B). A total of 390 DEGs were enriched by GO in the ck vs. gw3d group, among which the obsolete oxidation-reduction process, transmembrane transport, oxidoreductase activity, metabolic process, and other BPs were significantly enriched. MF, such as ATP binding, heme binding, flavin adenine dinucleotide binding, and response to stress, were enriched, as were CCs, such as the membrane, nucleus, and fungal-type cell wall (Figure 3C).

GO enrichment analysis of DEGs in *S. rugosoannulata* under drought stress indicated that 107 DEGs in ck vs. gh3d group were significantly enriched in CCs (including the membrane, nucleus, and fungal-type cell wall), MF (including ATP binding, protein binding, oxidoreductase activity, flavin adenine dinucleotide binding, and catalytic activity), and BPs (including obsolete oxidation-reduction processes, transmembrane transport, and metabolic processes) (Figure 4A). A total of 294 DEGs were enriched in the ck vs. gh6d group, and the DEGs were significantly enriched in obsolete oxidation-reduction processes, transmembrane transport, metabolic processes, protein kinase activity, response to stress, and other items in BPs. In CCs, the differential genes were significantly enriched in the membrane, nucleus, cytoplasm, and other items. In MF, differential genes were significantly enriched in iron ion binding, oxidoreductase activity, flavin adenine dinucleotide binding, protein binding, and transmembrane transporter activity (Figure 4B). In addition, 812 DEGs enriched by GO were found in the ck vs. gh9d. In terms of BPs, obsolete oxidation-reduction processes, transmembrane transport, metabolic processes, responses to stress, and other related DEGs were significantly enriched. As far as CCs are concerned, differential genes were significantly enriched in the membrane, nucleus, fungal-type cell wall, and cytoplasm. In terms of MF, ATP binding, protein binding, oxidoreductase activity, heme binding, iron ion binding, protein kinase activity, and other related DEGs were significantly enriched (Figure 4C).

### 3.5. KEGG Enrichment Analysis of Differentially Expressed Genes

Similarly, KEGG signaling pathway enrichment analysis was carried out on the DEGs screened between the group of high temperature and drought stress treatment versus the control group, and the top 20 most significantly enriched pathways were listed based on the number of enriched DEGs. The results revealed that the pathways enriched in the ck vs. gw1d group mainly included protein processing in endoplasmic reticulum (ko04141), amino sugar and nucleotide sugar metabolism (ko00520), DNA replication (ko03030), pyruvate metabolism (ko00620), purine metabolism (ko00230), and the MAPK signaling pathway (ko04011) (Figure 5A). The pathways enriched in the ck vs. gw2d group mainly include the MAPK signaling pathway (ko04011), cysteine and methionine metabolism (ko00270), pyruvate metabolism (ko00620), phenylalanine metabolism (ko00360), and cyanoamino acid metabolism (ko00460) (Figure 5B). In addition, the pathways enriched in the ck vs. gw3d group mainly include the MAPK signaling pathway (ko04011), protein processing in the endoplasmic reticulum (ko04141), pyruvate metabolism (ko00620), steroid biosynthesis (ko00100), and cysteine and methionine metabolism (ko00270) (Figure 5C).

KEGG pathway enrichment analysis was conducted on DEGs in *S. rugosoannulata* under different drought stress treatments, and the top 20 most significantly enriched pathways were listed based on the number of enriched DEGs. The results showed that the pathways enriched in the ck vs. gh3d group mainly include pentose and glucuronate interconversions (ko00040), butanoate metabolism (ko00650), alanine, aspartate, and glutamate metabolism (ko00250), spliceosome (ko03040), and the MAPK signaling pathway (ko04011) (Figure 6A). The pathways enriched in the ck vs. gh6d group mainly include the MAPK signaling pathway (ko04011), steroid biosynthesis (ko00100), pentose and glucuronate interconversions (ko00040), meiosis (ko04113), and cysteine and methionine metabolism (ko00270) (Figure 6B). In addition, the pathways enriched in the ck vs. gh9d group mainly include the MAPK signaling pathway (ko04011), cell cycle (ko04111), meiosis (ko04113), DNA replication (ko03030), and nucleotide excision repair (ko03420). The results revealed that DEGs in *S. rugosoannulata* may adapt to drought stress through dynamic regulation of different pathways (Figure 6C).

In conclusion, the MAPK signaling pathway (ko04011) was significantly enriched in the DEGs between all stress treatment groups and the control group, indicating that this signaling pathway may play an important regulatory role in response to high temperature and drought stress in *S. rugosoanulata*. Therefore, DEGs in this pathway can be used as candidate genes for stress responses in subsequent gene annotation analyses.

### 3.6. Screening of Critical Candidate Genes’ Response to High Temperature and Drought

In order to explore candidate genes that may be involved in stress resistance regulation of *S. rugosoannulata*, differential genes annotated in the KEGG pathway (ko04011) enriched in all treatment and control groups were selected for further analysis. The results showed that four, 13, and 27 genes were annotated in ck vs. gh3d, ck vs. gh6d, and ck vs. gh9d, respectively. Among them, two genes (*TRINITY_DN164_c0_g1*, *ATF2*, up-regulated; and *TRINITY_DN351_c0_g1*, *PTP2_3*, up-regulated) responded simultaneously among all groups, and 10 genes were repeatedly annotated in ck vs. gh6d and ck vs. gh9d (Appendix A). In addition, four, 11, and 12 genes were annotated in ck vs. gw1d, ck vs. gw2d, and ck vs. gw3d under high-temperature stress, respectively. Among them, just one gene (*TRINITY_DN164_c0_g1*, *ATF2*, up-regulated) responded simultaneously among all groups, and eight genes were repeatedly annotated in ck vs. gw2d and ck vs. gw3d (Appendix A). In summary, these 15 annotated candidate genes (Appendix A) play a crucial role in response to high temperature and drought stress in *S. rugosoannulata*, providing genetically supported evidence for further analysis of the regulatory mechanism.

### 3.7. qRT-PCR Expression Analysis of Candidate Genes

In order to verify the accuracy of RNA-seq data and the expression pattern of candidate genes under high temperature and drought stress, eight candidate genes were randomly selected for qRT-PCR. As shown in Figure 7, the expression patterns of each candidate gene were differentially expressed under different stress treatments. Specifically, candidate genes *PTP2_3*, *BCK1*, *SHO1*, *GPCR*, and *SAC* were significantly expressed at 1 d of high-temperature stress, *ATF2* was significantly expressed at 2 d of high-temperature stress, while *GPA* and *PRMT5* were significantly expressed at 5 d of high-temperature stress. In addition, *BCK1*, *SHO1*, and *GPCR* were significantly expressed at 3 d of drought stress; *PTP2_3* and *PRMT5* were significantly expressed at 6 d of drought stress; while *ATF2*, *SAC*, and *GPA* were significantly expressed at 9 d of drought stress. The qRT-PCR results were basically consistent with the transcriptome sequencing data, which proved the reliability of the transcriptome sequencing data. In conclusion, the differential expression of each candidate gene under different stress treatments may be due to its response to high temperature and drought stress through different defense mechanisms in different regulatory pathways. These findings will provide important gene resources for stress-resistance breeding of *S. rugosoannulata*.

## 4. Discussion

*S. rugosoannulata* is an important edible and medicinal fungus, which is delicious and nutritious and has health care and medicinal values such as enhancing human immunity and lowering blood sugar [2,34,35,36]. *S. rugosoannulata* belongs to medium-low temperature edible fungi, and its mycelium growth temperature ranges from 5 °C to 35 °C, and the temperature range of fruiting body formation and growth is from 10 °C to 20 °C [10,37]. In cultivation practice, the mycelium is easy to burn when the feed temperature is above 30 °C, and when the temperature rises to 20 °C, the fruiting body is easy to open, the quality becomes worse, and the primordium does not differentiate at 25 °C, which leads to cultivation failure or yield reduction [38,39,40]. Up to now, there are few reports on screening candidate genes of *S. rugosoannulata* responding to high temperature and drought stress by the high-throughput sequencing technique. In this study, transcriptome sequencing, GO functional classification, KEGG metabolic pathway analysis, and gene annotation were used to explore the critical candidate gene of *S. rugosoannulata* under stress to provide research reference for obtaining stress-tolerant strains in the future.

When organisms are exposed to high temperature, drought, and other adverse conditions for a long time, it will cause cell dehydration, an increase in reactive oxygen species (ROS), and intracellular level disorder [12,41,42]. Whether it is edible fungi or animal and plant breeding, the aim is to obtain new varieties with excellent characteristics. The high temperature and drought tolerance of edible fungi is a concrete manifestation of its extensive adaptation to the environment, which is closely linked with high yield. Hence, the screening of resistant strains is also one of the main characteristics of high-yield traits in *S. rugosoannulata* [11,40]. Therefore, high temperature and drought environments will not only reduce the metabolic activity of mycelium and affect the growth of mycelium but also cause damage to mycelium and reduce economic benefits. In order to cultivate varieties with stronger resistance to high temperature and drought stress, here, we used transcriptome analysis to study the *S*. *rugosoannulata* strain D-3 to better understand its expression pattern and response mechanism under high temperature and drought stress. Chen et al. [43] compared the mycelium growth of the collected *Auricularia auricula* at different growth temperatures for different treatment times and preliminarily identified three *A. auricula* varieties with high temperature resistance. In the preliminary experiment of this study, we found that long-term high temperature and drought stimulation could easily exceed the tolerance limit of *S. rugosoannulata* mycelia, resulting in degradation or even death of mycelia. Even if the suitable growth environment were restored, the growth rate and growth of mycelia would also deteriorate. In addition, 15 candidate genes of *S. rugosoannulata* in response to high temperature and drought stress were finally identified in this study. It is worth noting that one candidate gene was repeatedly identified in all treatment groups. The excavation of these genes is of great significance to improve the regulation mechanism of *S. rugosoannulata* in response to high temperature and drought stress and provides gene resources for its stress resistance breeding in the future.

Transcriptome sequencing technology, as an important research technique for identifying DEGs and screening stress-resistant genes, can not only help researchers quickly and globally understand all transcriptional changes in plant response to stress but also accurately obtain the target transcript sequence and the information of expression level, thereby quickly and accurately screening stress-related genes. Ma et al. [44] carried out transcriptomic analysis using drought- or high-temperature-resistant seedlings, and the results indicated that 11,731 and 9639 DEGs were identified, including 4444 drought-induced genes and 7287 drought-inhibited genes, and 5493 significantly up-regulated genes and 4146 significantly down-regulated genes under high-temperature stress treatment, respectively. Meanwhile, the critical genes that respond to both drought and high-temperature stress were identified, such as *WRKY*, *HSF*, and *MYB*. A report by Ren et al. [10] indicated that the catalase (*CAT*) and superoxide dismutase (*SOD*) activity of the thermotolerant *S. rugosoannulata* strain L3 were higher than those of the heat-sensitive *S. rugosoannulata* strain SM. The expression patterns of trehalose-6-phosphate phosphatase (*TPP*), *SOD*, and *CAT* in the thermotolerant strain L3 were higher than those in the heat-sensitive strain SM. Furthermore, Minina et al. [45] revealed that the up-regulated expression of the *atg5* and *atg8* mutant was closely related to the antioxidant response. Under drought stress, the significantly up-regulated expression of *atg5* and *atg8* in *Morchella sextelata* may affect the stress response of cells, resulting in increased autophagy, which may enhance the resistance to drought stress, thereby promoting the viability of *M. sextelata* in harsh environments. This study identified 15 candidate genes responding to drought stress through RNA-seq technology in *S. rugosoannulata*. The results in the present study will be useful in expanding our insight into drought and high-temperature stress biology. Notably, two candidate genes were repeatedly identified among all treatment groups. The isolation, functional research, and study on the molecular mechanism of stress resistance of these candidate genes will be the focus of future research.

Under drought stress, *app1* in fungi can regulate the expression of important antioxidant enzymes and reduce oxidative stress caused by drought. In addition, it is also related to the steady-state regulation of copper ions, which further affects the redox balance of cells, thus playing a protective role under drought stress [46]. In addition, *PRO1* can regulate multiple signaling pathways under drought stress, including cell wall integrity, NADPH oxidase, and pheromone signal transduction, and plays an important role in the growth and development of fungi [47]. Moreover, Xin et al. [48] speculated that the expression of the hydrophobin *hyd1* gene could enhance the ability of *L. edodes* to resist high-temperature stress by using transcriptome technology. In summary, in combination with the 15 candidate genes screened in this study, no repeatedly reported candidate genes were found. This may be due to the different degrees of stress treatments of different edible fungi or different strains of *S*. *rugosoannulata*, resulting in different molecular mechanisms for responding to stress treatment. At present, there are few reports on the molecular mechanisms of *S*. *rugosoannulata* under stress treatment. However, with the rapid development of sequencing technology and molecular biology technology, elucidating the molecular mechanisms of its response to abiotic and biotic stresses will become a top priority for researchers. At present, the assembled genome of the *S*. *rugosoannulata* strain QGU27 (GCA_028532985.1) was posted in the NCBI database in 2023 (https://www.ncbi.nlm.nih.gov/datasets/genome/GCA_028532985.1/, accessed on 21 September 2025); however, there is no *S. rugosoannulata* strain D-3 genome information published, and there is no report on the molecular response mechanism of *S. rugosoannulata* to high temperature and drought stress at the transcriptome level, and the genetic information available for reference is insufficient, which limits the mining of *S. rugosoannulata* functional genes to a certain extent. In this study, based on RNA-seq technology, the DEGs of *S. rugosoannulata* in response to high temperature and drought stress were preliminarily screened, and the expression patterns of eight candidate genes were verified. However, the in-depth analysis and functional identification of these genes are worthy of further work.

## 5. Conclusions

Our study provides the first comprehensive discovery of the transcriptome of *S*. *rugosoannulata* under drought and high-temperature stress. We identified 74,571 transcripts and 16,233 unigenes, out of which 7506 DEGs were differentially regulated. GO analysis showed that the DEGs were mainly involved in biological processes such as the obsolete oxidation-reduction process, transmembrane transport, and response to stress. In addition, the MAPK signaling pathway (ko04011) was significantly enriched in all the stress treatment groups versus the control group. Further, the present study identifies 15 candidate genes that respond to both high temperature and drought stress through annotation. The expression levels of eight selected candidate genes were carried out via qRT-PCR, and the results were consistent with the transcriptome datasets, indicating the data’s reliability. Until now, this is the first study to screen stress-related candidate genes by RNA-seq in *S*. *rugosoannulata* under high temperature and drought stress conditions. Therefore, our study provides valuable information and gene resources for the molecular breeding and the regulation mechanism for drought and high temperature resistance in *S. rugosoannulata* and other edible fungi species.

## Figures and Tables

**Figure 1 cimb-47-00835-f001:**
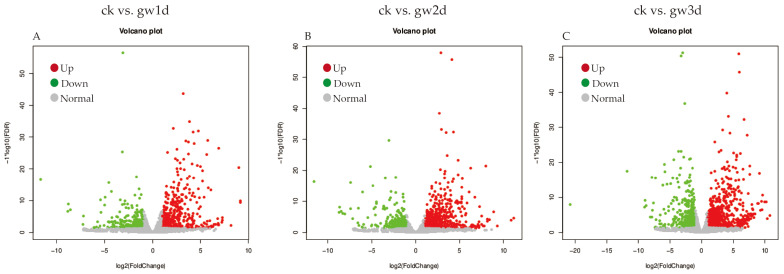
DEG analysis between the high-temperature stress treatment and the control group. Volcano plot analysis of DEGs for the pairwise comparisons ck vs. gw1d (**A**), ck vs. gw2d (**B**), and ck vs. gw3d (**C**).

**Figure 2 cimb-47-00835-f002:**
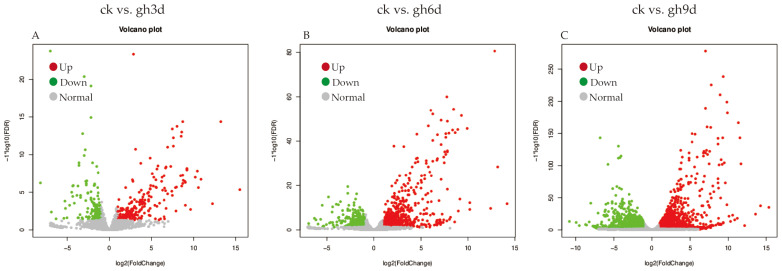
DEG analysis between the drought stress treatment and the control group. Volcano plot analysis of DEGs for the pairwise comparisons ck vs. gh3d (**A**), ck vs. gh6d (**B**), and ck vs. gh9d (**C**).

**Figure 3 cimb-47-00835-f003:**
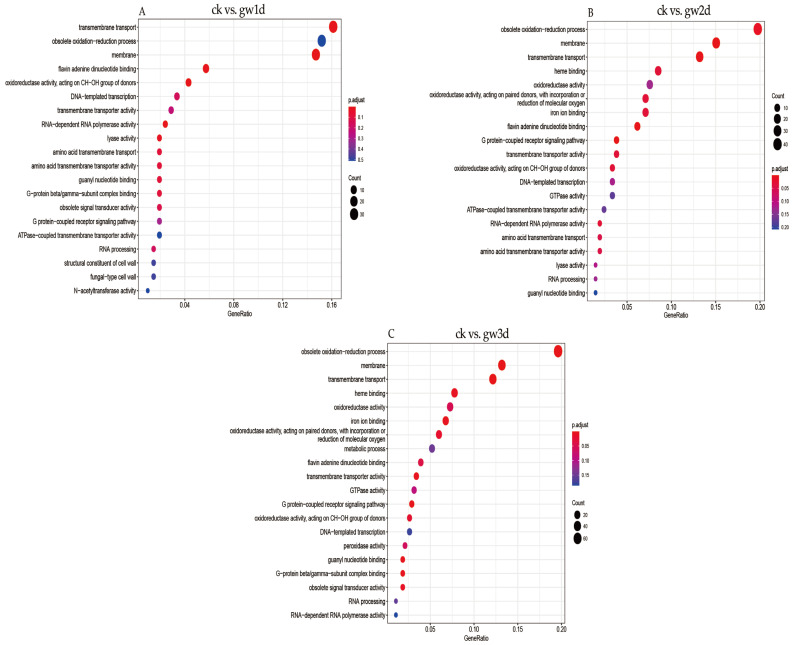
GO enrichment analysis of DEGs. Enrichment analysis of DEGs for the pairwise comparisons ck vs. gw1d (**A**), ck vs. gw2d (**B**), and ck vs. gw3d (**C**).

**Figure 4 cimb-47-00835-f004:**
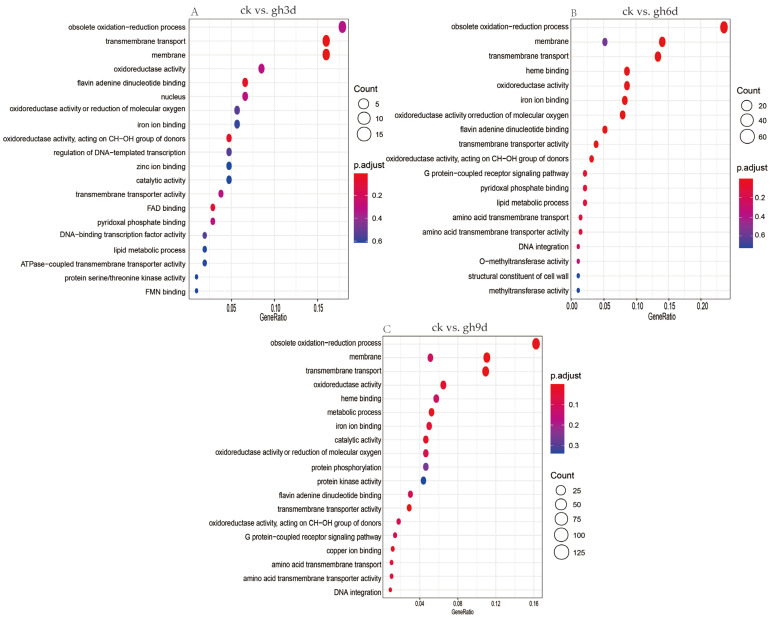
GO enrichment analysis of DEGs. Enrichment analysis of DEGs for the pairwise comparisons ck vs. gh3d (**A**), ck vs. gh6d (**B**), and ck vs. gh9d (**C**).

**Figure 5 cimb-47-00835-f005:**
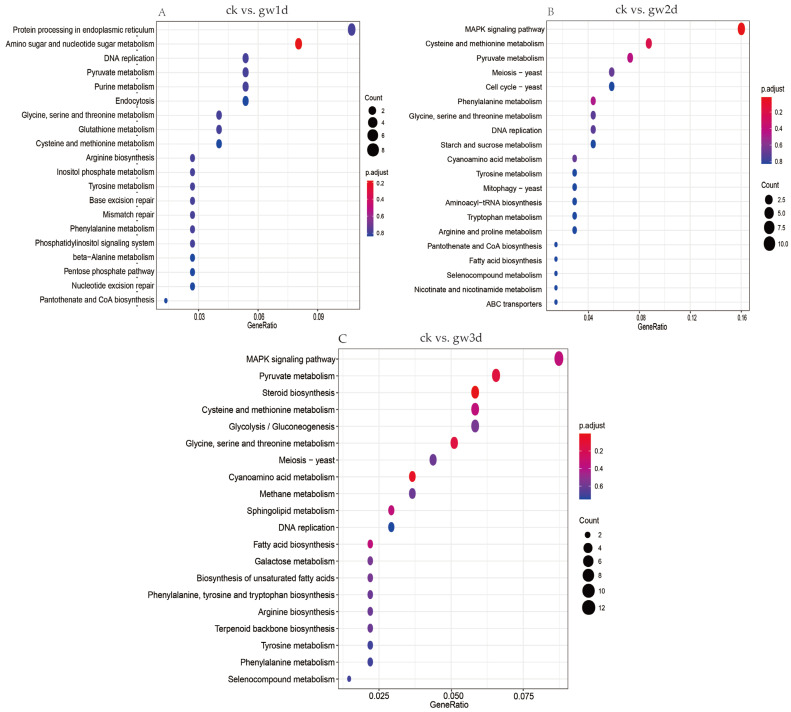
KEGG enrichment analysis of DEGs. Enrichment analysis of DEGs for the pairwise comparisons ck vs. gw1d (**A**), ck vs. gw2d (**B**), and ck vs. gw3d (**C**). The longitudinal axis represents the name of the pathway, and the size of the bubble represents the number of DEGs in this pathway.

**Figure 6 cimb-47-00835-f006:**
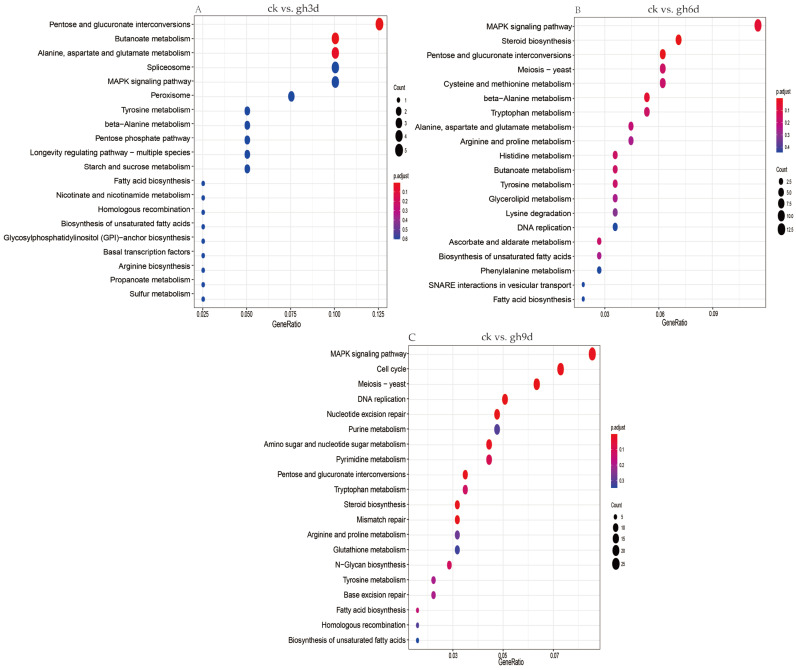
KEGG enrichment analysis of DEGs. Enrichment analysis of DEGs for the pairwise comparisons ck vs. gh3d (**A**), ck vs. gh6d (**B**), and ck vs. gh9d (**C**). The longitudinal axis represents the name of the pathway, and the size of the bubble represents the number of DEGs in this pathway.

**Figure 7 cimb-47-00835-f007:**
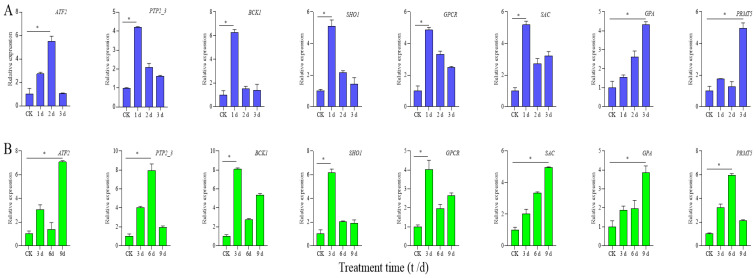
Relative expression levels of eight selected candidate genes were analyzed using qRT-PCR. (**A**) The expression patterns of eight candidate genes under high-temperature stress at different time points in *S. rugosoannulata*; (**B**) the expression patterns of eight candidate genes under drought stress at different time points in *S. rugosoannulata*. The x-axes show different stress treatments, and the y-axes show the relative expression level of each selected gene. Bars represent the standard deviation (SD) (n = 3) of three technical replicates. * Duncan’s new multiple range test indicates that candidate genes are significantly differentially expressed at the level of *p* < 0.05.

**Table 1 cimb-47-00835-t001:** Summary of RNA-seq of all samples.

Items	Raw Reads ^a^ (M)	Clean Reads ^b^ (G)	The Average Clean Data (G) ^c^	Q30 ^d^
gh3d_1	45.06	6.74	6.78	97.99
gh3d_2	46.19	6.91		98.00
gh3d_3	44.78	6.7		97.49
gh6d_1	44.54	6.67	6.58	97.96
gh6d_2	45.42	6.8		98.02
gh6d_3	41.96	6.28		98.25
gh9d_1	46.12	6.9	7.12	97.90
gh9d_2	48.04	7.19		97.8
gh9d_3	48.54	7.26		97.84
gw1d_1	45.24	6.77	6.83	98.05
gw1d_2	47.52	7.11		98.02
gw1d_3	44.12	6.6		97.77
gw2d_1	43.54	6.52	6.36	97.88
gw2d_2	40.78	6.09		98.02
gw2d_3	43.32	6.48		98.12
gw3d_1	43.35	6.49	6.83	98.07
gw3d_2	49.15	7.36		97.80
gw3d_3	44.45	6.65		98.00
ck_1	45.29	6.77	6.97	97.74
ck_2	47.95	7.18		97.77
ck_3	46.50	6.96		98.03
Total	951.86	142.43	-	-

a~d represent the raw reads number of sequencing data; the number of clean reads; the average of clean reads per stress treatment sample; and the percentage of bases with a quality value greater than or equal to 30 to the total number of bases, respectively.

## Data Availability

The original contributions presented in this study are included in the article/Appendix A. Further inquiries can be directed to the corresponding author(s).

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
