# Peer review of "Transcriptome Analysis Reveals Critical Genes Involved in the Response of Stropharia rugosoannulata to High Temperature and Drought Stress"

_cimb, 2025, doi:10.3390/cimb47100835_

Round 1

Reviewer 1 Report

Comments and Suggestions for Authors

This manuscript presents the results of transcriptome analysis of the response of the mushroom Stropharia rugosoannulata to two types of stress factors – drought and high temperature. The authors identified genes whose expression significantly changed in response to stresses of different durations and which, therefore, can be part of the mechanism regulating the mushroom resistance to these stressors. Analysis of the DEGs identified 15 candidate genes that most significantly responded to drought or high temperature. The work is interesting and relevant in terms of breeding new stress-resistant mushroom strains.

The Abstract clearly states the results of the study. However, the first sentence seems unfinished – it should be completed or rephrased.

The introduction provides detailed information about the S. rugosoannulata (of medicinal importance), its growing conditions, including special requirements for temperature and humidity, and the effect of violation of these conditions on the morphology of the mushroom.

Lines 39-46: “It was found that the crude protein content of S. rugosoannulata was ... are 81.5 mg/100 g and 53.1 mg/100 g, respectively [6].” It seems to me that the text given here is not needed, since this information does not intersect with the purpose and results of the study. Well, may be except that the content of flavonoids/phenols and vitamin C affects stress resistance.

Lines 88-102: Studies by other authors using RNA-seq of other fungal species in response to various abiotic stresses are mentioned, where a large number of DEGs were identified. More specific information is needed here – which specific groups of genes or individual genes were identified in those studies as candidates for participation in the regulation of fungal stress tolerance.

In the Material and Methods:

Lines 144-146: “Because there is no reference genome of S. rugosoannulata D-3, we use Trinity v2.15.1 [28] to splice clean reads to get transcripts, and then 145 carry out subsequent analysis.” The assembled genome of the Stropharia rugosoannulata strain QGU27 (genome assembly ASM2853298v1) was posted in the NCBI database in 2023 (Feb 7, https://www.ncbi.nlm.nih.gov/datasets/genome/GCA_028532985.1/). Please comment on this in the text.

Line 160: Please provide correct links instead of this address: https://www.ncbi.nlm.nih.gov/pub/COG/KOG/kyva

In the Results:

L203: “gh3d, gh6d, gh9d, gw1d, gw2d, and gw3d”. Please complete this list with a description of what is what, e.g. response to drought (...), response to high temperature (...).

L265-… (P. 7-11): “…DEG were enriched… In terms of cellular component (CC), membrane, nucleus, fungal-type cell wall, and other related DEG were significantly enriched…” and so on. It would be more correct to say that not DEGs are enriched in something, but certain GO/KEGG terms are enriched in DEGs. In addition, it is not clear from the description how many DEGs (of the found 212, 214, 390… DEGs) were upregulated and how many DEGs were downregulated in response to stress. It is also unclear how the lists of DEGs overlapped by time points (how many DEGs were repeated, and how many new ones arose between, for example, 3 and 6 days of drought, and whether they retained the direction of expression change (up or down)). Venn diagrams are often used effectively in articles to visualize such data.

para.3.6: Please clarify here whether these candidate genes are upregulated or downregulated. Also, it would be useful to supplement Tables S4, S5, and S6 with the names of genes and proteins that they presumably encode. Homologues found using NCBI-blast-x in annotated data of various fungal species (closest to S. rugosoannulata) can be used for protein names. In Table S7, it is necessary to specify for which organism the 15 genes were annotated and to provide the database (NCBI?) identification numbers of these homologues.

In Discussion:

Please discuss how the obtained results are consistent with those available in the literature cited in the discussion: do they overlap, are they new, did you find similar genes, what can be said if you combine the presented and already published materials, can you suggest some mechanisms for regulating stress resistance, etc.

Lines 467: “At present, there is no S. rugosoannulata genome information published.” The assembled genome of the Stropharia rugosoannulata strain QGU27 (genome assembly ASM2853298v1) was posted in the NCBI database in 2023 (Feb 7, https://www.ncbi.nlm.nih.gov/datasets/genome/GCA_028532985.1/). Please, comment.

Author Response

Response to Reviewer 1 Comments

Dear reviewer,

We are glad to receive your valuable comments and suggestions to our manuscript. Thank you for your kind consideration on this manuscript "Transcriptome Analysis Reveals Critical Genes Involved in the Response of Stropharia rugosoannulata to High Temperature and Drought Stress". Without your professional reviews, this manuscript would not be as smooth and more persuasive as what it is now. Thank you very much!

We have amended the manuscript according to all the opinions, suggestions and comments of the reviewers and all the changes have been marked-up in the text by the red fond. The responses to all the comments and suggestions are itemized as follows:

 Comment 1: The Abstract clearly states the results of the study. However, the first sentence seems unfinished – it should be completed or rephrased.

Response 1: Thanks for your professional comment. This sentence has been rephrased.

Comment 2: The introduction provides detailed information about the S. rugosoannulata (of medicinal importance), its growing conditions, including special requirements for temperature and humidity, and the effect of violation of these conditions on the morphology of the mushroom.

Response 2: Thanks for your professional comment. S. rugosoannulata is a rare edible mushroom that has been promoted for cultivation both domestically and internationally in recent years. It is rich in protein, dietary fiber, and minerals such as potassium and selenium, which help enhance immunity and also provide good economic benefits.

Comment 3: Lines 39-46: “It was found that the crude protein content of S. rugosoannulata was ... are 81.5 mg/100 g and 53.1 mg/100 g, respectively [6].” It seems to me that the text given here is not needed, since this information does not intersect with the purpose and results of the study. Well, may be except that the content of flavonoids/phenols and vitamin C affects stress resistance.

Response 3: Thanks for your professional comment. This sentence has been deleted.

Comment 4: Lines 88-102: Studies by other authors using RNA-seq of other fungal species in response to various abiotic stresses are mentioned, where a large number of DEGs were identified. More specific information is needed here – which specific groups of genes or individual genes were identified in those studies as candidates for participation in the regulation of fungal stress tolerance.

Response 4: Thanks for your professional comment. The corresponding specific information has been added.

Comment 5: Lines 144-146: “Because there is no reference genome of S. rugosoannulata D-3, we use Trinity v2.15.1 to splice clean reads to get transcripts, and then carry out subsequent analysis.” The assembled genome of the Stropharia rugosoannulata strain QGU27 (genome assembly ASM2853298v1) was posted in the NCBI database in 2023 (Feb 7, https://www.ncbi.nlm.nih.gov/datasets/genome/GCA_028532985.1/). Please comment on this in the text.

Response 5: Thanks for your professional comment. We have added the corresponding genome information you mentioned.

Comment 6: Line 160: Please provide correct links instead of this address: https://www.ncbi.nlm.nih.gov/pub/COG/KOG/kyva

Response 6: Thanks for your professional comment. We have made corrections in the manuscript.

Comment 7: L203: “gh3d, gh6d, gh9d, gw1d, gw2d, and gw3d”. Please complete this list with a description of what is what, e.g. response to drought (...), response to high temperature (...).

Response 7: Thanks for your professional comment. We have added the corresponding description you mentioned.

Comment 8: L265-… (P. 7-11): “…DEG were enriched… In terms of cellular component (CC), membrane, nucleus, fungal-type cell wall, and other related DEG were significantly enriched…” and so on. It would be more correct to say that not DEGs are enriched in something, but certain GO/KEGG terms are enriched in DEGs. In addition, it is not clear from the description how many DEGs (of the found 212, 214, 390… DEGs) were upregulated and how many DEGs were downregulated in response to stress. It is also unclear how the lists of DEGs overlapped by time points (how many DEGs were repeated, and how many new ones arose between, for example, 3 and 6 days of drought, and whether they retained the direction of expression change (up or down)). Venn diagrams are often used effectively in articles to visualize such data.

Response 8: Thanks for your professional comment. In fact, in part 3.3, we made an analysis of the up-regulated and down-regulated of DEGs. In order to avoid content duplication, if possible, we will contact the editor to submit the corresponding Supplementary Materials in the submission system or manuscript.

Comment 9: para.3.6: Please clarify here whether these candidate genes are upregulated or downregulated. Also, it would be useful to supplement Tables S4, S5, and S6 with the names of genes and proteins that they presumably encode. Homologues found using NCBI-blast-x in annotated data of various fungal species (closest to S. rugosoannulata) can be used for protein names. In Table S7, it is necessary to specify for which organism the 15 genes were annotated and to provide the database (NCBI?) identification numbers of these homologues.

Response 9: Thanks for your professional comment. We have added the contents in Table S4-7 according to your requirements. In addition, we will contact the editor to submit another supplementary material (the ORF sequences of 15 candidate genes) either through the submission system, in the manuscript, or directly via email.

Comment 10: Please discuss how the obtained results are consistent with those available in the literature cited in the discussion: do they overlap, are they new, did you find similar genes, what can be said if you combine the presented and already published materials, can you suggest some mechanisms for regulating stress resistance, etc.

Response 10: Thanks for your professional suggestion. According to your suggestions, we combined previous research reports with the results of this study, and added some discussion content in the manuscript.

Comment 11: Lines 467: “At present, there is no S. rugosoannulata genome information published.” The assembled genome of the Stropharia rugosoannulata strain QGU27 (genome assembly ASM2853298v1) was posted in the NCBI database in 2023 (Feb 7, https://www.ncbi.nlm.nih.gov/datasets/genome/GCA_028532985.1/). Please, comment

Response 11: Thanks for your professional comment. We have added the corresponding genome information you mentioned.

Take this opportunity, again we do want to express our high and great appreciation for your kind words and professional suggestions, which should no doubt help for our future research. Thanks again.

Any questions, we will be more than happy to answer. Looking forward to hearing from you soon.

Regards and Best wishes!

Reviewer 2 Report

Comments and Suggestions for Authors

The authors leveraged transcriptomics to investigate the genes and pathways that regulate the response against heat and drought stress in Stropharia rugosoannulata. The results are clearly presented, essentially supporting the authors’ conclusions. Overall, I believe this manuscript is solid and could be published. I only have a few minor suggestions/questions:

It would be appreciated if the authors could use the same format (p value, count, etc) for Figure 3 and 4.

Some of the pathways shown in Figure 5 and 6 are annotated ‘yeast’ or even ‘multiple species’. Is it possible to cluster the results to be more fungi-exclusive?

Comments on the Quality of English Language

Puctionations are not properly used in some sentences (the first sentence in Abstract, for instance). Please double check.

Author Response

Response to Reviewer 2 Comments

Dear reviewer,

We are glad to receive your valuable comments and suggestions to our manuscript. Thank you for your kind consideration on this manuscript "Transcriptome Analysis Reveals Critical Genes Involved in the Response of Stropharia rugosoannulata to High Temperature and Drought Stress". Without your professional reviews, this manuscript would not be as smooth and more persuasive as what it is now. Thank you very much!

We have amended the manuscript according to all the opinions, suggestions and comments of the reviewers and all the changes have been marked-up in the text by the red fond. The responses to all the comments and suggestions are itemized as follows:

Comment 1: It would be appreciated if the authors could use the same format (p value, count, etc) for Figure 3 and 4.

Response 1: Thanks for your professional comment.Firstly, we found that there is some missing information in Figure 3, so we recombined the figure. Then, in the GO enrichment analysis, the clusterProfiler function is used to conduct enrichment analysis on differentially expressed genes (DEGs), calculating p-values, which are then adjusted for false discovery rate (FDR) to control the false positive rate. The adjusted p-values (p-adjust) are more conservative but also more reliable. In the GO enrichment analysis, the differences in p-values between different samples are mainly due to the following factors: 1. The number of DEGs varies, which can significantly change the p-value. For example, if sample A has a background gene count of 1,000 and sample B has a background gene count of 5000, even if the enrichment levels are the same, sample B will have a lower p-value. 2. Differences in enrichment levels; the p-value reflects the significance level of the enrichment. If sample A is enriched with 10 genes in a specific GO term while sample B is only enriched with 5 genes, even if the background gene counts are the same, sample A's p-value will be smaller (more significant). Therefore, to obtain the most significant and reliable enrichment results, we cannot use the same p-value and count for enrichment analysis.

Comment 2: Some of the pathways shown in Figure 5 and 6 are annotated ‘yeast’ or even ‘multiple species’. Is it possible to cluster the results to be more fungi-exclusive?

Response 2: Thanks for your professional comment. In the KEGG enrichment analysis, the top 20 most significantly enriched pathways were listed according to the number of differentially expressed genes (DEGs) that were enriched. In this study, the enriched pathways for 'yeast' or even 'multiple species' do not seem to provide an intuitive graphical representation, and they do not cluster completely within the same branch. We are very sorry and hope to gain your understanding and approval.

Comment 3: Puctionations are not properly used in some sentences (the first sentence in Abstract, for instance). Please double check.

Response 3: Thanks for your professional suggestion. All the authors have double checked the full text many times. and all the changes have been marked-up in the text by the red fond.

Take this opportunity, again we do want to express our high and great appreciation for your kind words and professional suggestions, which should no doubt help for our future research. Thanks again.

Any questions, we will be more than happy to answer. Looking forward to hearing from you soon.

Regards and Best wishes!